# Cell Growth Model with Stochastic Gene Expression Helps Understand the Growth Advantage of Metabolic Exchange and Auxotrophy

Dibyendu Dutta,[a] Supreet Saini[a]

aDepartment of Chemical Engineering, Indian Institute of Technology Bombay, Mumbai, India

**ABSTRACT** During cooperative growth, microbes often experience higher fitness by sharing resources via metabolite exchange. How competitive species evolve to cooperate is, however, not known. Moreover, existing models (based on optimization of steady-state resources or fluxes) are often unable to explain the growth advantage for the cooperating species, even for simple reciprocally cross-feeding auxotrophic pairs. We present here an abstract model of cell growth that considers the stochastic burst-like gene expression of biosynthetic pathways of limiting biomass precursor metabolites and directly connect the amount of metabolite produced to cell growth and division, using a "metabolic sizer/adder" rule. Our model recapitulates Monod's law and yields the experimentally observed right-skewed long-tailed distribution of cell doubling times. The model further predicts the growth effect of secretion and uptake of metabolites by linking it to changes in the internal metabolite levels. The model also explains why auxotrophs may grow faster when supplied with the metabolite they cannot produce and why two reciprocally cross-feeding auxotrophs can grow faster than prototrophs. Overall, our framework allows us to predict the growth effect of metabolic interactions in independent microbes and microbial communities, setting up the stage to study the evolution of these interactions.

**IMPORTANCE** Cooperative behaviors are highly prevalent in the wild, but their evolution is not understood. Metabolic flux models can demonstrate the viability of metabolic exchange as cooperative interactions, but steady-state growth models cannot explain why cooperators grow faster. We present a stochastic model that connects growth to the cell's internal metabolite levels and quantifies the growth effect of metabolite exchange and auxotrophy. We show that a reduction in gene expression noise can explain why cells that import metabolites or become auxotrophs can grow faster and why reciprocal cross-feeding of metabolites between complementary auxotrophs allows them to grow faster. Furthermore, our framework can simulate the growth of interacting cells, which will enable us to understand the possible trajectories of the evolution of cooperation *in silico*.

**KEYWORDS** cross-feeding, mutualism, cooperation, microbial communities, stochastic growth model, metabolite exchange, Adder, Sizer, gene expression noise, division of labor, resource allocation

In the laboratory, we study microbes in isolation; however, different species live side by side in the environment, either competing for the limited resources or cooperating with each other. Defying general evolutionary expectations, cooperative communities pervade different ecological niches (1–5), highlighting its purported selective advantage (6). Cooperative interactions often involve secretions, ranging from extracellular enzymes, scavenging molecules (like siderophores) (7), to leaked metabolites—either toxic by-products (8, 9) or essential metabolites like amino acids (10–12).

Address correspondence to Dibyendu Dutta, dibyendu_dutta@iitb.ac.in.

How do the loss of genes and the exchange of metabolites enable cooperators to grow faster? Our model finds that auxotrophy and direct metabolite import reduces the contribution of gene expression noise, enabling faster growth.

Cooperation may also involve living together in a biofilm, which protects the group from toxins, antibiotics, and predators (13).

Understanding the advantages of cooperation in ecological communities (14) and implementing them in synthetic communities have been challenging (15–17). The focus has been on metabolic modeling to identify the keystone "currencies" of cooperation (9, 18–23). The identities of the "traded" metabolites help uncover the mechanisms of cooperation in different environmental conditions. However, to perdure in an ecological niche, cooperative consortiums must demonstrate high community productivity and growth compared to other competing (free-living or cooperating) species. While metabolic modeling approaches can determine the metabolic viability and synergy in terms of the cooperative consortium's productivity of metabolic flux, they usually do not yield the growth kinetics of the consortium. Hence, studying the evolution of cooperation using existing evolutionary frameworks like fitness landscapes has not been possible.

Furthermore, there are several shortcomings in the existing modeling approaches. Microbial growth is often posed as an optimal resource allocation problem and solved assuming that the cell operates at a steady state. However, computing the steady state itself often uses an estimate of the growth rate as the rate of dilution due to cell division, which leads to inaccurate estimates of growth. Often such models consider only a part of the cell and coarse grain its dynamics to maximize the energy generation while minimizing the resource investment in catabolic enzymes and transporters (24, 25). More sophisticated whole-cell models like bacterial growth laws account for the shortcomings by computing a proxy measure of growth rate using the steady-state ribosome fraction of the proteome (26–29). Metabolic flux models assume a biomass objective function and only compute the steady-state optimal genome-wide flux contributions to biomass to uncover the pathways or enzyme reactions that bottleneck growth (30–34).

These steady-state approaches neglect the cell-to-cell differences due to stochastic gene transcription (35–37), leading to differences in the numbers of proteins (38) and the cell's metabolic state (39). While such variations are unimportant for predicting the mean characteristics of the population, it is critical to model the growth of a cooperative consortium with cell-to-cell interactions such as the exchange of metabolites.

In this work, we develop a model that captures the effect of stochastic variation on the growth of bacterial cells that may or may not exchange metabolites. We implement stochastic burst-like gene transcription using Monte Carlo methods following the framework in Golding et al. (40). To determine when a cell divides, we combined the concept of a biomass objective function (41) with the empirical laws for bacterial cell growth and size homeostasis: Adder and Sizer (42–45). While Adder proposes that cells divide upon adding a fixed length to the birth size, Sizer proposes that division occurs when the cell grows to a characteristic length. We postulate that any cell size increment corresponds to an equivalent quantity and stoichiometric composition of biomass precursor metabolites. Thus, the cell divides only after producing the required quantity of metabolites in the correct stoichiometry.

Our framework attempts to capture and study the dynamics of cooperative growth when cells exchange metabolites. We wanted to find the factors leading to the faster growth of the consortium to investigate their evolutionary relationships. We obtained the distribution of cell doubling times, which mimics the experimentally observed distributions (46, 47) obtained from single-cell growth experiments run on microfluidic devices (42, 48). Our model recapitulates Monod's hyperbolic relationship between the substrate and growth rate (49). Furthermore, we demonstrate that gene expression noise manifests as noise in metabolite levels that delay cell division (i.e., the noise is time additive). We also demonstrate that reducing the effective number of limiting metabolites in a cell, either by auxotrophy or by the direct import of the metabolite, can accelerate growth. Last, we demonstrate that the growth rate of reciprocal cross-feeding auxotrophs can be greater than the prototrophs. In all, by incorporating

stochastic gene expression noise into a cellular growth model, we capture the growth effect of auxotrophy and metabolite exchange directly and show how it may help understand the evolution of cooperation.

## RESULTS

**Model development.** We develop a microbial growth model that avoids relying on cellular steady states and incorporates the stochastic variation in growth.

As discussed, recent studies have put forth the empirical size-based laws for bacterial cell division: Adder and Sizer (42–45). Combining these with the idea of biomass objective functions (41), we argue that cell size increase requires producing an equivalent quantity of constituent metabolites in proper stoichiometry. We propose the idea as the "metabolic adder" and "metabolic sizer" and use it as the foundation of our proposed cell growth model. Cell division is triggered only when the cell produces the minimum amounts of all the necessary metabolites (Fig. 1a). Our model recapitulates many known cellular phenomena and can even describe the kinetics of single-cell growth while cells exchange metabolites among each other. We find the Adder and Sizer models very similar in their outcomes and proceed with Sizer for all the rest of the simulations in this work, primarily because it is found to be the better fit for the outcomes in case of poor medium conditions, where growth is on minimal media with single substrates (44) (see Supplement S4 at https://bit.ly/32ayGpb).

Now, while theoretically, a shortfall in any metabolite could prevent a cell from dividing, only a few metabolites at a time may act as "kinetic" bottlenecks to cell division (i.e., waiting for the production of which delays cell division). These may be required in substantial amounts, or the respective enzyme's production may be variable and in low numbers (39), such that the metabolite demand cannot be met in time. Additionally, some metabolites may be prone to leakage (may serve as public goods) and hence may act as a bottleneck to growth (50). We assume that the cell is comprised of $p$ such anabolic pathways, whose products bottleneck cell division. All metabolite biosynthesis (anabolic) pathways are fed by the substrate supplied by the upstream catabolic pathways via different shunts from the central carbon metabolism. We simplify the arrangement and assume that all anabolic pathways are fed a common substrate supplied at a constant rate $S_{in}$ per second per bottleneck metabolite (in total $p \times S_{in}$ per second) (Fig. 1b) (see Supplement S10 at https://bit.ly/32ayGpb for unequal bottlenecks).

The distribution of cell doubling times across different growth conditions collapse to one distribution when scaled by their means (47), which leads us to postulate that the noise in common downstream metabolic pathways like anabolism may be responsible for the observed noise in growth, since under different growth conditions, upstream catabolic pathways may differ. Hence, our modeling approach focuses on quantifying the noise from anabolic pathways alone (see Supplement S11 at https://bit.ly/32ayGpb for catabolic noise). Furthermore, to circumvent the differences due to the exact structure and kinetic properties of different anabolic pathways, we assume all $p$ pathways to be linear and comprised of $n$ enzyme-catalyzed steps, all of which we assume have identical kinetic properties and transcription parameters.

We further assume that these $n$ enzymes in each pathway are present on the same operon, and hence, the $n$ enzymes share the same transcriptional burst profile (Fig. 1c). To incorporate the effect of stochasticity in our model, we implement stochastic burst-like gene expression, using the findings of exponentially distributed durations of transcription burst ($t_{ON}$) and the waiting time between successive bursts ($t_{OFF}$) (40). We randomly sample the ($t_{OFF}$) and ($t_{ON}$) durations iteratively from two exponential distributions of set means, to generate the stochastic burst profile (Fig. 1c). Next, assuming no shortage of RNA polymerases, transcription proceeds at a constant rate until the end of $t_{ON}$, where any incomplete transcripts are terminated. Each mRNA produced is assigned a lifetime by sampling from an exponential distribution. During the lifetime of an mRNA, only a single ribosome attaches and translates proteins. We assume that

mSystems®

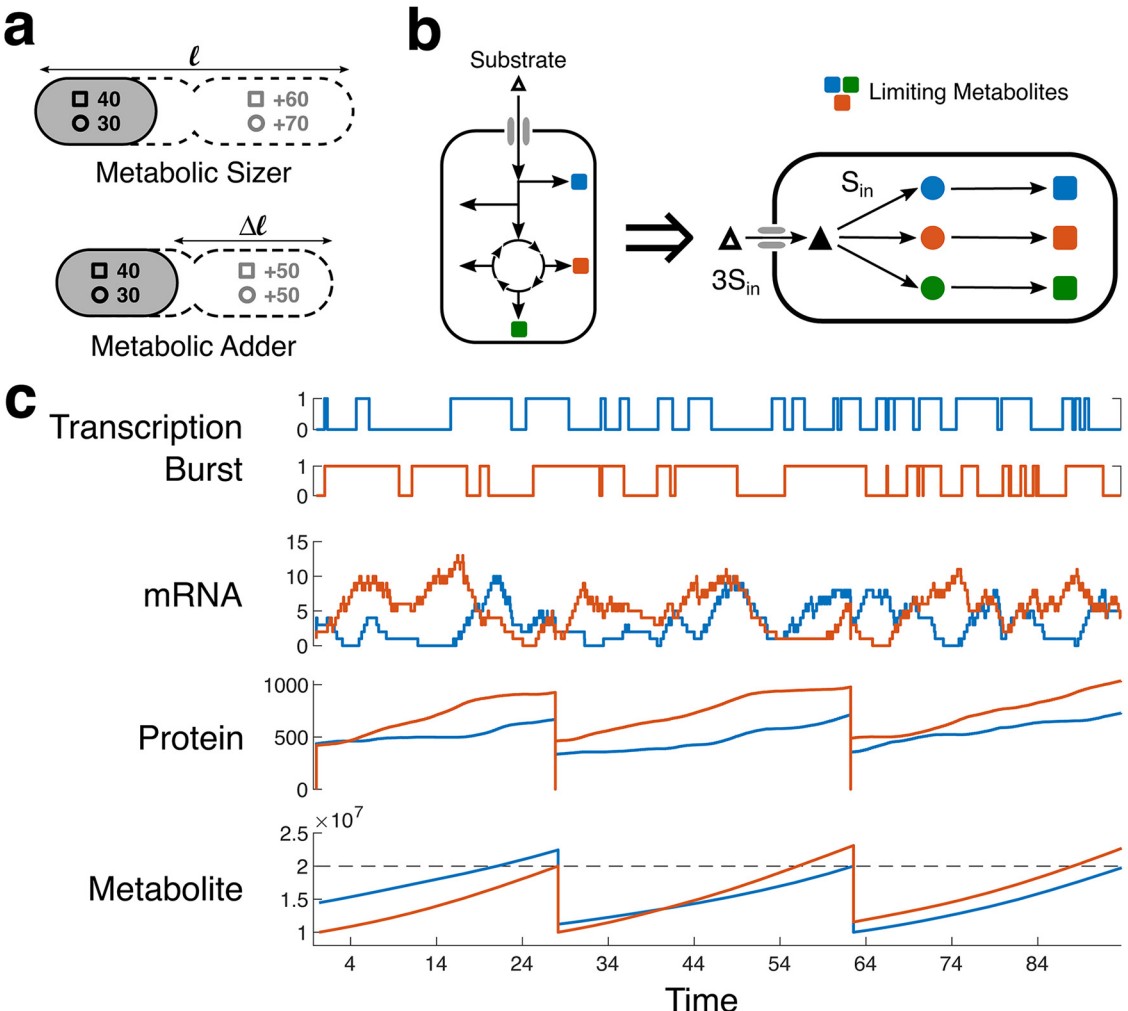

**FIG 1** Model schematic and behavior of internal variables. (a) Model schematic. The bacterium takes up the substrate molecules from the environment and processes it via metabolic pathways to produce all the necessary metabolites for survival. Some (*p*) metabolites (colored squares) bottleneck cell growth, since sufficient quantities cannot be produced in time. Our model assumes a simplified arrangement for all these limiting metabolite anabolic pathways as linear enzyme cascades with *n* enzyme steps, all fed by a common substrate supplied at a constant rate. We consider all enzymes are equivalent in terms of expression and kinetic properties and are expressed in stochastic bursts. ℓ, length. (b) Our model implements a "metabolite adder/sizer" model, wherein we reinterpret the empirical size-based Adder and Sizer laws, in terms of the metabolites necessary for cell size increase such that cell division is triggered upon production of these metabolites in the required amounts and stoichiometry: always a fixed amount in the case of Adder and the balance amount in the case of Sizer. (c) Considering our model for *p* = 2 limiting metabolites and the metabolic Sizer law. The model implements stochastic burst-like gene expression as a random process of transcription bursts switching on and off. During the burst, mRNA is produced, which acts as the template for protein translation. Both mRNA and proteins decay stochastically but mRNA decays much faster. The various protein enzymes allow the production of the required metabolites at various rates. According to Sizer, the cell divides when both the metabolites cross the threshold (dashed line). After division, cellular contents are halved.

ribosomes are always sufficient (a ribosome limitation is expected to reduce the production of proteins but not affect the overall dynamics). Any incomplete protein is terminated once the mRNA decays. Each protein produced is assigned a lifetime by sampling from an exponential distribution.

We use coupled ordinary differential equations (ODEs) based on Michaelis-Menten kinetics for the metabolite production, with the stochastic protein profiles as time-varying parameters (see Materials and Methods). We convert all quantities to molecules per cell assuming a cellular volume of 1 $\mu$m³.

All the biosynthetic pathways compete for the common substrate, and the flux they can acquire depends only on the number of enzymes present at each time since kinetic

parameters are assumed equal. When all the $p$ metabolites cross their requirement threshold based on the "metabolic adder or sizer" law, the cell divides. After division, all the cellular content is divided between the two daughter cells equally (51). Thus, the daughter cells start from the inherited values of the mRNA, proteins, and metabolites but generate unique stochastic burst profiles, going forward.

**Model properties and outcome.** Simulating our model for a given set of parameters yields the time between successive cell divisions or the cell generation time. The histogram of the generation times for all simulated parameter sets reveals a right-skewed distribution with a long tail, which concurs well with many experimental observations of bacterial growth at a single-cell resolution (42, 44, 45, 47, 52, 53) (see Supplement S2 at https://bit.ly/32ayGpb for our analysis of the data set in reference 42). These studies indicated that the observed generation time distribution resembled a Gaussian on a logarithmic scale and hence may be a gamma or log normal distribution (47). Pugatch, on the other hand, analyzed the single-cell data set extensively and made a distinction between the quantities: cell generation time ($T_{div}$) (or interdivision time) and the cell doubling time ($T_\mu$), the time for the doubling of cell size (46).

$$\text{Doubling time}\,(T_\mu) = \frac{\text{Generation time}\,(T_{div})}{\log_2\left(\frac{\text{cell size at division}}{\text{cell size at birth}}\right)}$$

He tested the fit of different distributions to the experimentally derived doubling time distribution and found that a log Fréchet or log generalized extreme value (GEV) (type II) distribution provided the best fit to the data.

In our framework, cell division is triggered by exact size laws and division yields perfect halves (see Supplement S6 at https://bit.ly/32ayGpb for alternate modeling criteria). Hence, the ratio of size at division and birth is always 2, and thus, generation time and doubling time are the same. We tested various combinations of gene expression and metabolite threshold parameters to obtain distributions that resemble physiology (see Supplement S3 at https://bit.ly/32ayGpb). The simulated generation times obtained using the selected parameters give a long-tailed right-skewed distribution which is fit best by a log generalized extreme value (GEV) (type III) distribution (shape parameter $k < 0$) (Fig. 2a) (see supplemental Fig. S3c for comparison of GEV distribution shapes [https://bit.ly/32ayGpb]). Moreover, since the generation time distribution obtained is the maximum of the production times of the $p$ metabolites, the obtained GEV-like distributions match with intuition derived from the extreme value theory (as the limit distribution of the maxima of a sequence of independent and identically distributed random variables) (54).

Now, if the number of bottleneck pathways ($p$) increases, the chance that stochastic variations in enzymes would prevent the cell from reaching production thresholds at a given time increases. Hence, the maximum of the first passage times from $p$ pathways, i.e., the generation time increases. The distribution of cell generation times shifts to the right and reduces skewness (Fig. 2b; supplemental Fig. S1a at https://bit.ly/32ayGpb). Moreover, upon scaling the data with the mean, we find that the distributions overlap (Fig. 2b, inset), but with increasing $p$, the coefficient of variation (CV) of the distribution increases, while skewness and kurtosis decreases (supplemental Fig. S1a at https://bit.ly/32ayGpb).

Substrate flux per pathway ($S_{in}$) represents the available concentration of the nutrients in the environment. The downstream anabolic pathways utilize the substrate for metabolite production. When $S_{in}$ is low, the rate of metabolite production is low, and cell generation times are longer, but as $S_{in}$ increases, metabolite production picks up, and cell generation times are shorter, and we observe that the distribution of cell generation times shifts to the left (Fig. 2c). Upon scaling the data with the mean, the distributions overlap (Fig. 2c, inset), but with increasing $S_{in}$, skewness and kurtosis decrease, and distribution CV also increases and saturates (see supplemental Fig. S1b at https://bit.ly/32ayGpb). Next, by plotting the associated population growth rates against the corresponding $S_{in}$ values, we observe a hyperbolic relationship for all

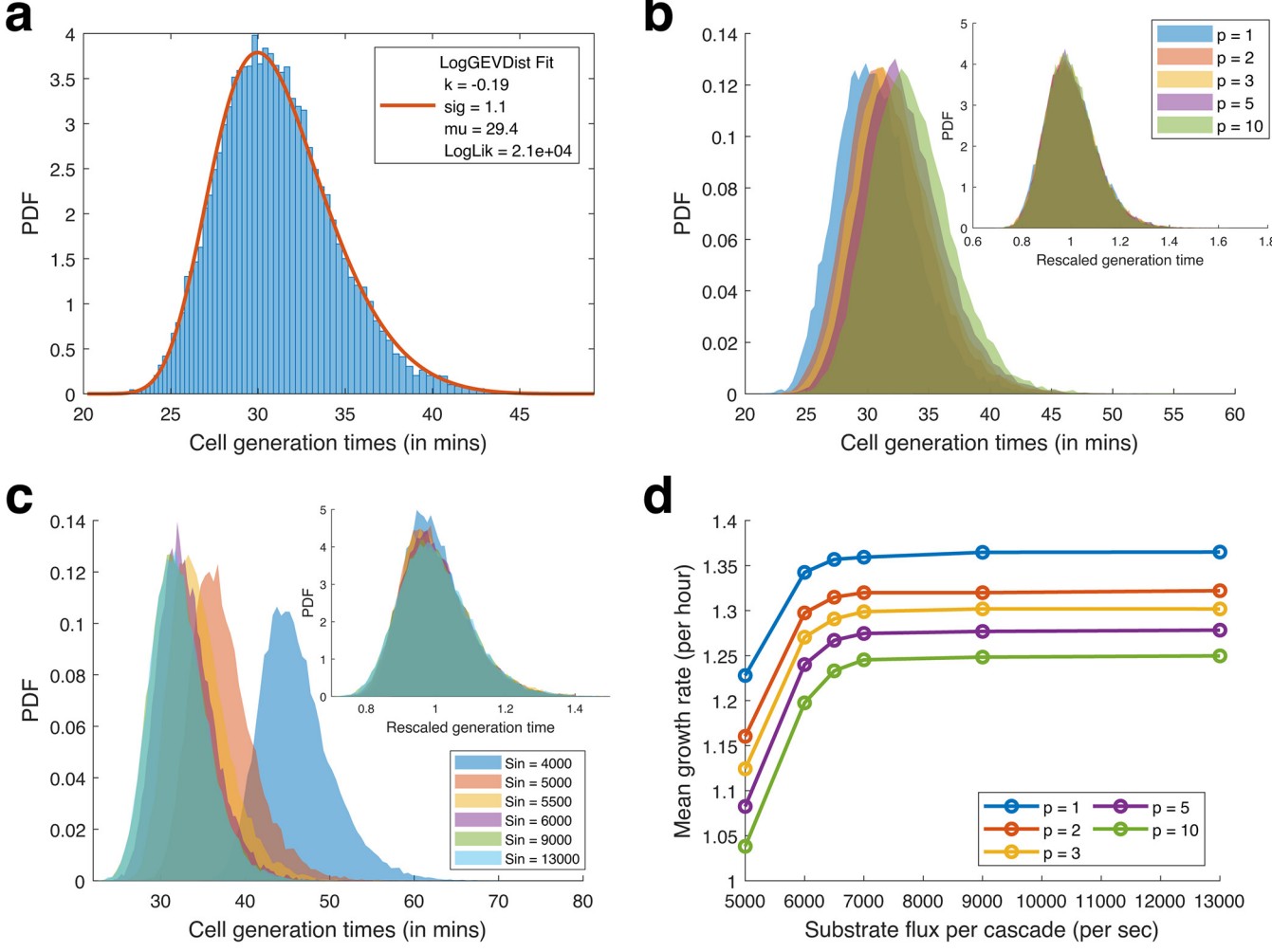

**FIG 2** Model properties. (a) Distribution of cell generation times obtained from model simulation is fitted best by log GEV distributions. The distribution for simulation with $p = 3$ and $S_{in} = 9,000$/s is shown. (b) Comparison of distribution of generation times for various numbers of limiting metabolites ($p$) at a high rate of substrate flux ($S_{in} = 9,000$/s). The inset shows the distributions rescaled to their means. (c) Comparison of distribution of generation times varying the substrate flux rate ($S_{in}$) in the simulations for the case of $p = 3$ metabolite bottlenecks. The inset shows the distributions rescaled to their means. (d) Comparison of the mean growth rate of the simulated cells with various numbers of bottleneck metabolites ($p$) for different rates of substrate flux ($S_{in}$). The hyperbolic relationship is reminiscent of Monod's law.

simulated values of $p$, which recapitulates the hyperbolic relationship between substrate concentration and the cell's growth rate observed by Monod (49) (Fig. 2d).

Substrate flux is known to be the primary determinant of growth rate in a cell, and hence, studies have focused on quantifying the noise in catabolic pathways as a measure of substrate flux variation to estimate growth rate variations (36). However, we intend to study how growth is affected when the cell imports and secretes different biomass precursor metabolites or loses a biosynthetic pathway. Therefore, we focus on the anabolic pathways that synthesize these metabolites instead of the common catabolic pathways that provide the substrate flux to all of the pathways.

Studies have shown that cell division is controlled by more than one factor and limiting agent (55, 56), which in our model is represented by multiple bottleneck pathways ($p$). In a real cell, however, the number of bottlenecks may depend on the cellular growth conditions, and in the case of poor medium conditions (low $S_{in}$), it may increase. Although our model cannot perform such internal metabolic switches, it can still recapitulate physiological observations such as the distribution of generation times and Monod's law (Fig. 2d).

**Effect of metabolite uptake and secretion on simulated cells.** Our primary goal is to develop a framework that can quantify the effect of nutrient uptake and secretion

on the growth of cells. Our model design captures the effect of importing external metabolites and exporting internally produced metabolites directly in terms of the generation time.

Till now, our simulations have considered that cells were grown on media that provides only the substrate flux ($S_{in}$). We now simulate growth when additional limiting metabolites are present in the media (like amino acids, nucleotides). Our model cell imports these metabolites directly at a constant rate to meet the internal metabolite requirement. The imported metabolite effectively lowers the amount that needs to be produced internally (i.e., the metabolite threshold), and thus, cells on average divide faster. Since we do not explicitly model any metabolite feedback, the substrate flux availed by the biosynthetic pathway does not reduce, and the substrate competition remains the same. (We relax this condition in Supplement S12 at https://bit.ly/32ayGpb.) With increasing import rates of the limiting metabolite, the generation times shift to the left (Fig. 3a and b), and the growth rate increases (Fig. 3c). From our model's perspective, importing a limiting metabolite at a high rate makes it nonlimiting. Thus, decreasing the number of bottlenecks ($p \rightarrow p - 1$) removes its noise contribution to cell division (Fig. 4d), allowing the cell to divide faster. The growth advantage from importing a limiting metabolite has an upper bound, and the advantage from a higher rate of import saturates quickly (Fig. 3d).

Next, we consider the effect of exporting metabolites from the cell. Some metabolites are usually produced in excess and leak out from the cell (10–12, 50) without any adverse effect on growth. However, if these metabolites are not useless by-products for the cell, then excess secretion could impede growth. In case one of the limiting metabolites is secreted, it leads to an effective increase in its metabolite threshold, and hence, the cell requires more time to divide. We quantify the growth effect of secreting metabolites by simulating secretion as a fraction of the total metabolite produced per unit time, rather than a constant amount secreted per unit time since it synchronizes the secretion to the internal production. It hence captures an accurate estimate of the amount secreted (low secretion after birth due to low enzyme and higher secretion before division due to a higher number of enzymes and transporters). As the secretion ratio increases, cell generation times increase, and the distribution shifts to the right (Fig. 3e and f). The magnitude of the difference is, however, much more pronounced in the case of nutrient-poor media (low $S_{in}$ [Fig. 3e]). With the distributions scaled to the means, we further see that increased secretion leads to lowered skewness. Moreover, skewness decreases more in nutrient-poor media (supplemental Fig. S1d at https://bit.ly/32ayGpb). Unlike the case with nutrient uptake, the effect of secretion is unbounded, and increased secretion further lowers growth rate (Fig. 3g and h).

**Growth advantage of auxotrophy.** We have already demonstrated how the direct import of a limiting metabolite can increase growth and how the advantage is capped. We described this phenomenon as the partial alleviation of the noise contribution from one of the bottleneck metabolites, which corresponds to ($p \rightarrow p - 1$) in our model. If the modeled prototrophic cell mutates to lose one of the biosynthetic pathways for a limiting metabolite (and becomes an auxotroph), then the same outcome may be obtained. Interestingly, this is a common phenomenon seen in endosymbiotic bacteria (57) and some free-living bacteria (58, 59). The Black Queen Hypothesis proposes that this genome reduction is driven by a resulting selective advantage (50, 60). However, such gene loss makes the cell's survival contingent on a sufficient external supply of the metabolite.

For the simulation, we consider a cell with three bottleneck pathways ($p = 3$) and then delete one of them (Fig. 4a). The cell directly imports the metabolite it cannot synthesize at a constant "feed" rate. Additionally, the available internal substrate flux ($p \times S_{in}$), originally meant for $p$ pathways, is now redistributed among the remaining $p-1$ pathways. Thus, the auxotroph originally with three bottleneck pathways now receives 50% more substrate flux in the two remaining bottleneck pathways and hence divides faster (Fig. 4a). At low import (feed) rates, this missing metabolite determines

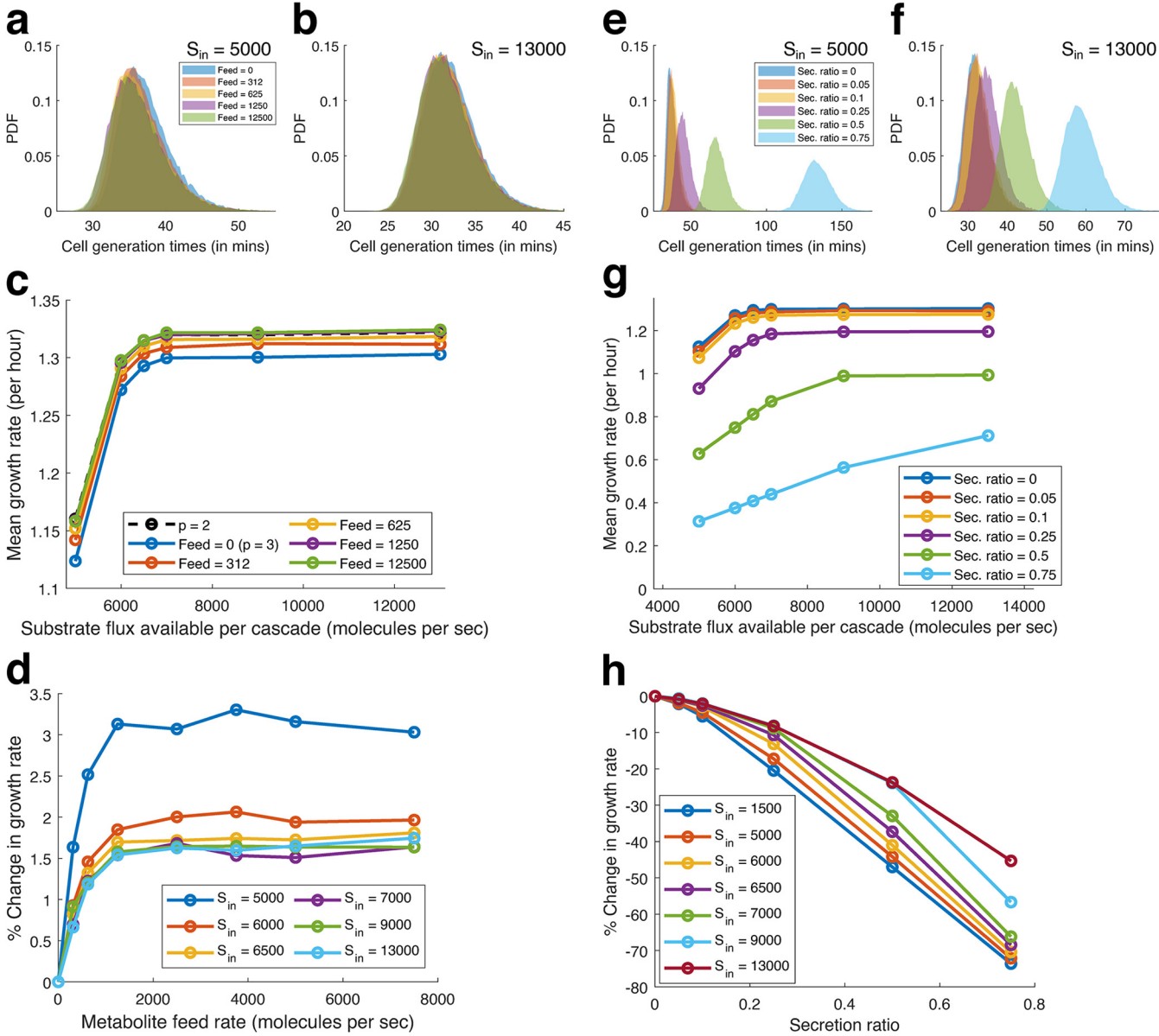

**FIG 3** Effect of metabolite uptake and secretion on simulated cells. (a to d) Effect of metabolite uptake. (a and b) Comparison of generation times for different rates of metabolite import, low substrate flux (a) and high substrate flux (b). (c) Comparison of the mean growth rate of the simulated cells with various rates of metabolite import, for different rates of substrate flux ($S_{in}$). (d) Comparison of the relative change in growth rate compared to the case of no uptake, due to various metabolite feed (uptake) rates, for different substrate flux ($S_{in}$). (e to h) Effect of metabolite secretion. (e and f) Comparison of generation times for different ratios of metabolite secretion (fraction of produced metabolite secreted), low substrate flux (e) and high substrate flux (f). (g) Comparison of the mean growth rate of the simulated cells with various metabolite secretion (Sec.) ratios for different rates of substrate flux ($S_{in}$). (h) Comparison of the relative change in growth rate compared to the case of no secretion, due to various metabolite secretion ratios, for different substrate flux ($S_{in}$).

the cell generation time; however, at higher import rates, the missing metabolite is no longer the bottleneck, and the generation time distributions overlap with the distribution for $p-1$ (i.e., $p = 2$) bottlenecks (data from Fig. 2d) (Supplement S8 at https://bit .ly/32ayGpb) as expected. It is interesting to note that when an auxotroph is grown in poor medium conditions, represented by the low substrate flux ($S_{in}$) in our simulations, a sufficient rate of import of the limiting metabolite allows the cells to grow much faster (Fig. 4b and c), relative to the growth advantage observed for high substrate flux conditions.

Our model does not incorporate the effect of saved protein costs. However, the effect of saved substrate flux is incorporated and reflected in the growth advantage

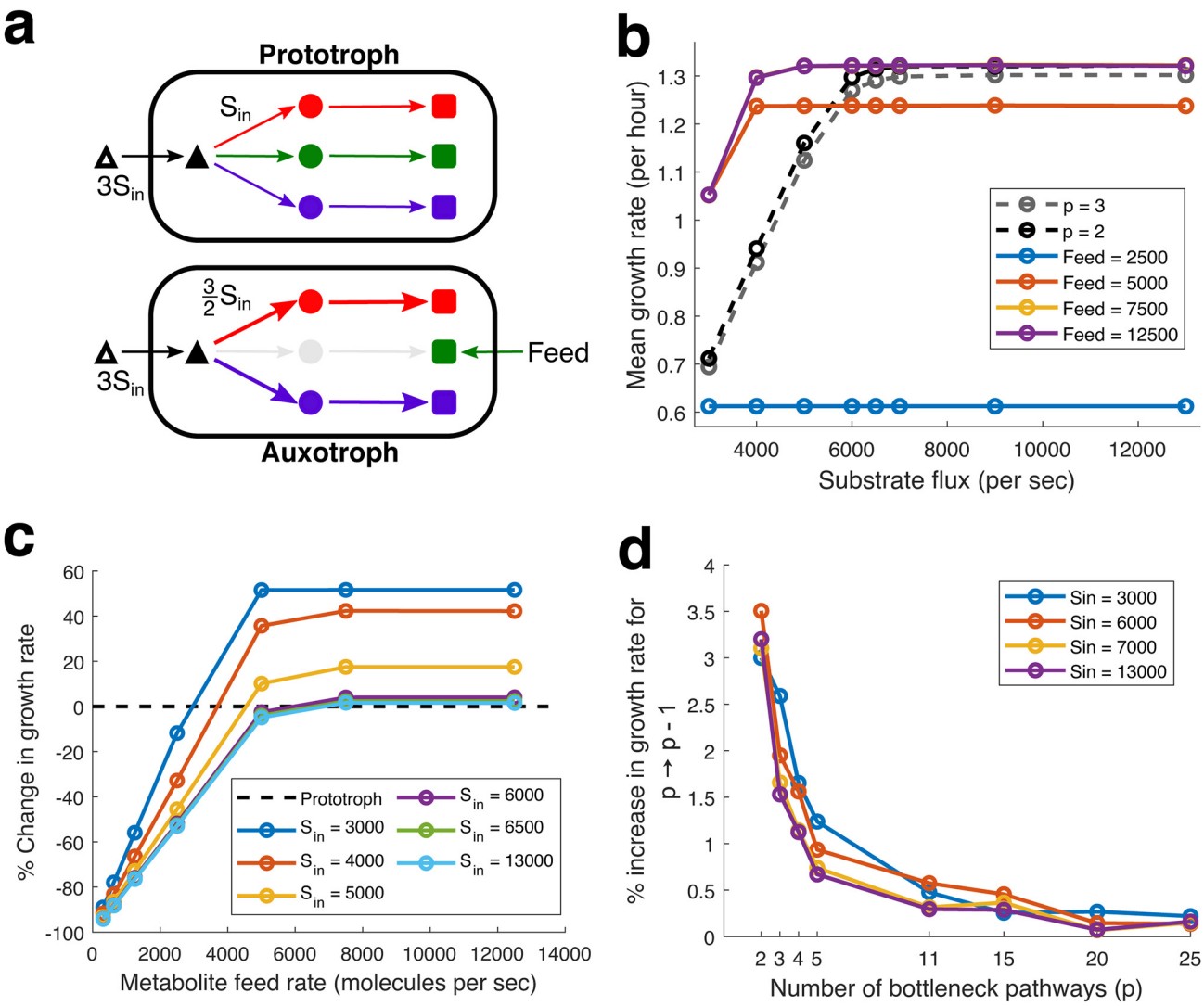

**FIG 4** Growth advantage of auxotrophy. (a) Schematic figure comparing an auxotroph ($p = 2$) with the ancestral prototroph ($p = 3$). (b) Comparison of the mean growth rate of the simulated auxotroph with various metabolite feed (uptake) rates for the missing metabolite (colored solid lines) for different rates of substrate import flux ($S_{in}$) on the x axis. The gray and black dashed lines represent the prototrophs with $p = 3$ and $p = 2$ bottleneck metabolites, respectively (data from Fig. 2d). (c) Comparison of the relative change in the growth rate of the auxotrophs for various rates of metabolite feed and substrate import flux ($S_{in}$) compared with the original prototroph ($p = 3$, grown without any additional metabolite feed). (d) Comparison of the percentage increase in the growth rate upon loss of a bottleneck pathway for different starting numbers of bottleneck pathways ($p$), simulated at different substrate import flux rates ($S_{in}$).

with a low substrate flux ($S_{in}$). Moreover, even when the effect of substrate flux saturates, the model predicts a growth difference between the ancestor prototroph and the auxotroph. Thus, our model presents a novel explanation contributing to the growth advantage due to gene loss, in addition to resource savings (in terms of protein cost and substrate flux) (50, 60), leading to the spontaneous evolution of auxotrophs (61).

It is imperative to note that the magnitude of growth advantage demonstrated here corresponds to a low number of bottlenecks (low $p$). If a large number of metabolites are bottlenecks simultaneously (large $p$), then the relative growth advantage observed due to loss of one pathway or direct import of one limiting metabolite will be significantly lower (Fig. 4d). While it is known that different metabolites act as bottlenecks for growth (62) for different nutrient media, it is not immediately clear which biomass precursor metabolites act as a bottleneck, how their numbers vary, and their

relative limiting effect on growth. In our model, we considered all bottlenecks of equal strength for simplicity (see Supplement S10 at https://bit.ly/32ayGpb for unequal bottlenecks).

**Towards reciprocal metabolic cross-feeding.** As discussed, the survival of auxotrophs is contingent on the external availability of the metabolite they cannot produce internally. Therefore, these cells invade ecological niches where other cells secrete the necessary metabolites, either as by-products or public goods. In a public-goods scenario, the invading cells act as "cheaters" and reduce community productivity (63–65). However, if the metabolite is a by-product and its accumulation is toxic, its clearance from the environment could induce the producer cells to secrete more and improve community productivity (8, 66, 67).

Analysis of available microbial genomes from various niches has revealed that most of the microbes are auxotrophic for at least a few essential metabolites, such as amino acids (60, 68). Although surprising, it explains why ecological examples of metabolite cross-feeding are commonplace (1–5). During the process of cross-feeding, the metabolites secreted by the microbes diversify the environment's nutrient composition, enabling survival and further evolutionary diversification of the group by allowing for the spontaneous evolution of auxotrophs (61).

However, prototrophic species could invade such cross-feeding populations by importing the secreted metabolites and "cheat" to grow faster and outcompete the consortium (69). To avoid such a fate, cross-feeding must also enable fast growth, enough to surpass the competing species, in addition to enabling the survival of the group. Nevertheless, we do not understand how the division of labor may allow cooperating cells to improve productivity enough to exhibit growth that is faster than prototrophs.

We have demonstrated that importing limiting metabolites accelerates growth, but the secretion of such limiting metabolites at high rates can also slow down growth. In a cross-feeding interaction, cells reciprocally exchange metabolites by secreting one metabolite and importing another simultaneously. How must cellular resources be balanced between these processes to achieve a net positive outcome is not completely understood, even for a simple symmetric exchange (although reference 70 provides some insights into the process).

Let us consider two reciprocal auxotrophs, each overexpressing the metabolite, the other needs for survival. These auxotrophs, devoid of one of the metabolic biosynthesis pathways, grow faster because they can redirect the saved resources from the pathway toward growth and reduce the effect of gene expression noise on growth (as we demonstrate using our model). However, the need to supply each other metabolites requires the cells to invest the saved resources toward metabolite overproduction, taking away the growth advantage. Assuming metabolites of similar costs are exchanged, intuitively, we may expect that the reinvestment of saved resources may enable a doubling of the metabolite production and hence allow the cells to exhibit a growth rate as high as an equivalent prototroph if we neglect the possible losses and delays due to transport of the metabolites or assume alternate arrangements like nanotubes (71). We demonstrate this result by simulating a steady-state version of our model (see Supplement S5 at https://bit.ly/32ayGpb). Thus, it is hard to imagine that cross-feeding can enable such growth advantages.

Pande et al. used synthetic reciprocal amino acid auxotrophs that also overproduced amino acids to study this phenomenon and found a large number of complementary auxotrophs, when cocultured could cross-feed stably, exhibiting growth rates faster than the parental prototroph in monoculture (11). However, only a few pairs could outcompete the prototroph in a competitive coculture, since the prototroph could feed on the secreted metabolites. Here, we use our stochastic growth framework in an attempt to explain these observations.

We start with an auxotroph ($p = 2$), originally a prototroph ($p = 3$) (Fig. 5a). Next, we implement the overexpression of the secreted metabolite. Since pathway enzymes are

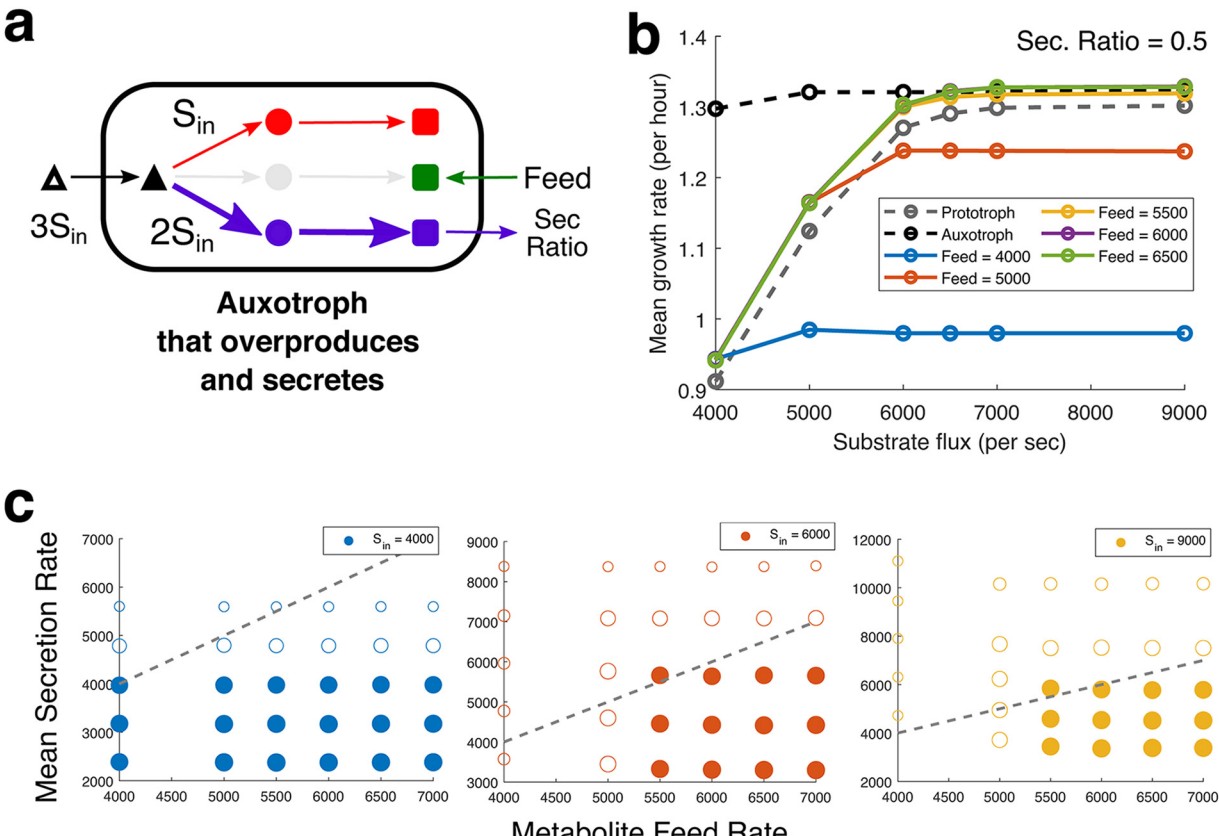

**FIG 5** Reciprocal metabolic cross-feeding. (a) Schematic figure depicting our proxy for reciprocal metabolic cross-feeding, an auxotroph that overexpresses and secretes one of the limiting metabolites (blue), while importing the metabolite it is unable to produce (green). Sec, secretion. (b) Comparison of the mean growth rate versus different rates of substrate import flux ($S_{in}$) for auxotrophs that overexpress (2×) and secrete (0.5×) a limiting metabolite (proxy for cross-feeders). The gray and black dashed lines represent the growth profile of corresponding prototroph (with no additional metabolite feed) and auxotroph (at saturating metabolite feed). (c) Comparison of the mean secretion rate of metabolites ($y$ axis) for an input metabolite feed rate ($x$ axis) for different substrate flux rates ($S_{in}$), represented as different colors of the plotted circles. The size of the circle represents the magnitude of the growth rate. The circles are filled if the simulated cells grow faster than the prototroph ($p = 3$) when grown without any additional metabolite feed. The dashed diagonal line shows when the secretion and feed rates are equal or identical.

equivalent in our model, the resources saved due to the loss of one pathway will allow another pathway to produce double the number of enzymes (maximum). However, it is unclear how real cells handle the overexpression of a pathway. In our stochastic framework, overexpression involves changing the transcription burst frequency, but since it would be hard to estimate burst parameters that precisely "double" the enzyme, we instead double the kinetic constant of the enzyme keeping the same stochastic burst profile to set up a fair comparison. Next, we considered secretion at different ratios of the amount of metabolite produced per unit time and also vary the direct import rate of the missing metabolite. Finally, we study the growth of this auxotroph that produces twice the quantity of metabolites per unit time while importing the missing metabolite as a proxy for the growth of cross-feeders (Fig. 5a).

We find that the proxy cross-feeders demonstrate growth rates higher than the prototroph, even when they secrete 50% of the metabolite they overproduce (Fig. 5b). For these overproducing and secreting auxotrophs to sustain a cross-feeding interaction, they need to secrete metabolites at a rate higher than or equal to the rate at which they import metabolites. We plot the mean secretion rates for each simulated metabolite feed (import) rate at different values of substrate import rate ($S_{in}$) (Fig. 5c). The dashed lines show that the secretion rate is equal to the metabolite feed. Thus, points above the diagonal represent secretion rates higher than the metabolite import. The

mSystems®

circle's radius represents the growth rate of the simulated cells, and the circle is filled if the growth rate is higher than the growth rate of the ancestor prototroph (without any metabolite feed). Thus, filled circles above the diagonal represent "feasible" parameters where cross-feeding cells can exhibit faster growth while maintaining higher overall metabolic productivity. In Fig. 5c, we find such feasible solutions at mid and high values of $S_{in}$. Thus, the kinetic advantage of cross-feeding consortiums may be observed only when a sufficiently concentrated substrate is available.

Overall, we demonstrate that feasible solutions for cross-feeding can be found even when there is minimal headroom for improvement of growth rate (i.e., in the symmetric reciprocal cross-feeding, where all the competing bottleneck pathways are of an equal threshold).

## DISCUSSION

We demonstrated that accounting for stochastic burst-like gene expression (40) captures variations in the production rates of "bottleneck" biomass precursor metabolites (39) and hence the differences in cell growth and division kinetics. All precursor metabolites considered in a genome-scale metabolic flux model's biomass objective function may bottleneck growth, but only at the optimal steady-state flux distribution. In our model, however, stochastic differences lead to variations in metabolite biosynthetic pathways. Metabolites with very high demand, or with many upstream flux shunts, or produced by biosynthetic enzymes with large gene expression noise are expected to be the "kinetic bottlenecks" as described in our model. However, due to a lack of suitable experimental evidence, we are currently unable to substantiate the existence and identity of these bottlenecks. By obtaining the changes in the experimentally observed distributions of bacterial cell generation times when they are fed different metabolites or are auxotrophic, and combining it with genome-scale metabolic models, we expect to identify them in the future.

Previously, Thomas et al. have developed a stochastic bacterial growth model that quantified the relative contributions of different sources of stochastic noise, in the growth rate (72), by building upon their previous steady-state model (26). They used the Cooper-Helmstetter model (44, 73) with Donachie's initiation constant (74) to determine when cells divide. At high growth rate, the model found that variations diminish; however, the variation in cell generation times persist (42). The model does not connect the stochastic production of individual biomass precursor metabolites and cellular growth and hence is unable to account for changes in growth upon their import or secretion.

Some studies have pitted genome-scale dynamic flux models of microbes with different auxotrophies and metabolite secretion rates against each other (75, 76). Some have additionally used evolutionary game theory to intuit evolutionarily successful microbial combinations (77, 78). These models, while useful in understanding the benefits of cooperation, cannot be used to simulate the population dynamics and the evolution of cooperation. Our framework, however, makes this possible by simulating the stochastic growth of individual cells in a population as they interact metabolically. Combined with a mutational framework, we expect our model will help uncover new insights toward the evolution of cooperation.

Our framework is however limited to predicting growth in a fixed medium condition. When the medium conditions shift drastically, the growth rates and associated Adder and Sizer lengths also change (42), requiring that the model's internal metabolite thresholds be updated.

Last, we acknowledge the assumption of a greatly simplified metabolic network in our model. Since all the enzymes are governed by the same kinetic and gene expression parameters, the common substrate is partitioned equally between the pathways on average. From the perspective of classic metabolic control theory (79), all the enzymatic steps in the metabolic network have equal flux coefficients and elasticities, and hence, each is a bottleneck of equal strength. Thus, growth effects observed through

our model arise exclusively from the consideration of stochastic variations in the enzymes.

## MATERIALS AND METHODS

**Model setup and calibration.** Our model ascribes no specific identity to the enzymes in the cell, and thus, we assume the simulated enzymes to have the average properties of all cellular enzymes. We obtain the average length of bacterial proteins, the median values of enzyme catalytic constant and enzyme half-saturation constant (see Supplement S9 at https://bit.ly/32ayGpb for a list of the parameters). For stochastic gene expression, we follow the scheme described by Golding et al. (40). We estimate parameters and the associated metabolite threshold by simulating the various combination of values and select for the mean generation time and coefficient of variation (CV) closest to the physiological values (see Supplement S3 at https://bit.ly/32ayGpb). We finally chose a metabolite threshold 1E7 for Adder, and 2E7 for Sizer, while $t_{ON}$ and $t_{OFF}$ chosen were 4 min and 2.4 min, respectively.

**Metabolic Adder and Metabolic Sizer.** Using the stoichiometric quantity of metabolites produced as a proxy for the cell size, we obtain the metabolic Adder and Sizer models. For simplicity, we assume 1E7 molecules of each limiting metabolite is required to make a new cell. Thus, for metabolic Adder, 1E7 more molecules of each limiting metabolite must be produced irrespective of the inherited metabolites. In the case of metabolic Sizer, the cell's total production of each limiting metabolite needs to exceed 2E7 to trigger cell division. Figure 1a shows this schematically.

**System of coupled differential equations used to compute metabolite production.** After generating the stochastic enzyme profile, we pass it as a time-varying parameter to a system of coupled ordinary differential equations (ODEs), based on Michaelis-Menten kinetics. However, instead of concentrations, all the variables are converted to absolute numbers assuming a cell of volume 1 $\mu$m³. See Supplement S9 at https://bit.ly/32ayGpb for a list of parameters used.

While it is possible to also consider the noise from the stochastic metabolic reaction processes, we disregard it in our model. The nature of this noise leads only to fluctuations in the mean catalytic constant of the enzyme, which becomes important at the timescales of single-molecule experiments (80), but averages out when we consider the metabolite production at the time scales of cell growth and division.

The primary substrate $S$ common to all the pathways is supplied at a constant rate $S_{in}$ per bottleneck metabolite (thus $p \times S_{in}$ in total), and the enzymes of the $p$ pathways feed off it. $i$ represents the bottleneck pathways 1 to $p$.

$$\frac{dS}{dt} = p \times S_{in} - \sum_{i=1}^{p} k_{cat}^{i1} E_{i1} \frac{S}{S + K_m^{i1}}$$

The subsequent products except the final products are modeled as:

$$\frac{dP_{i,j}}{dt} = k_{cat}^{ij} E_{ij} \frac{P_{i-1,j}}{P_{i-1,j} + K_m} - k_{cat}^{ij} E_{i+1,j} \frac{P_{i,j}}{P_{i,j} + K_m}$$

where $j$ represents the linear enzyme steps in each pathway.
The final step of the biosynthetic pathway is modeled as:

$$\frac{dP_{i,n}}{dt} = k_{cat}^{in} E_{in} \frac{P_{i,n}}{P_{i,n} + K_m}$$

For secretion, the production term is multiplied by (1 − secretion ratio). Constant import rate term is added for metabolite import. We considered each of the enzymes in the concurrent pathways equivalent in terms of their expression and kinetic parameters for our simulations. Thus, $k_{cat}^{ij} = k_{cat}$ for all $i.j$.

We evaluate the ODEs in MATLAB (81), using *ode15s* with an events function to trigger division when the metabolite threshold is reached. The number of enzymes $E_{ij}$ varies with time and is obtained by interpolation from the stochastic protein profile generated, every time the ODE function is evaluated.

**Model simulation.** We start the simulation of the model with the obtained parameters; however, we start with all zero for the initial conditions since we have no information about them. As we start the simulation, first, stochastic mRNA and protein profiles are generated based upon the set gene expression parameters, which are then used as time-dependent parameters while solving the system of coupled ODE numerically using *ode15s* in MATLAB (81). When all the $p$ metabolites have met their production requirements, the events function is triggered, which terminates the evaluation of the ODE, and marks the cell division. The difference between the end and start times of the ODE solution gives us the growth duration or generation time.

Before we start recording values from the simulation, we allow the system to stabilize and saturate by letting the cell run through 100 divisions or generations, following only one of the daughter cells, starting from the zero initial values. Thereafter, we simulate an exponential population growth by simulating all the daughter cells for 13 generations starting from 1 cell, and hence obtaining growth data for $2^{13} - 1 = 8,191$ cells. We use the generation times to compute the growth rate of the population, as described in the next section. We repeat the simulations with three or five unique seed values to obtain concordant observations.

**Computing growth rate from generation times.** From our simulations, we obtain the birth and division time of each simulated cell. If we sort the cells in the ascending order of their birth and consider each birth event as a unit increment to the population, it gives us the microbial population growth curve. The points appear as a straight line in a semilog graph; however, the last part appears to saturate because there are no more cell divisions. We consider only the first 40% of the sorted data set to obtain data from only the log or exponential growth phase to estimate the growth rate. We plot the best-fit line on the semilog plot and obtain the slope of the line as the population growth rate. To improve the estimate and make use of the entire data set, we take 100 permutations of the order of the cells during population growth and compute the growth rate. We take the average from the 100 estimates as the population growth rate.

**Fitting to log GEV distribution.** In a log GEV distribution, log-transformed variables follow a generalized extreme value (GEV) distribution. Thus, we plot the histogram of the log-transformed generation times and then fit a GEV distribution to the log-transformed data set to obtain the fitted probability density function (PDF). We note down the histogram bins' edges on the log scale and transform them back to the linear scale to obtain the log GEV distribution.

**Statistical tests.** We have performed two-way analysis of variance (ANOVA) tests using *anova2* in MATLAB (81) on the data sets comparing the effect of multiple parameters on the growth rate in the article to test whether the observed patterns are statistically significant. (See Supplement S7 at https://bit.ly/32ayGpb).

For all the tests, we use the raw simulated generation times. We have $2^{13} - 1 = 8,191$ data points for each independent run with a unique seed value (for the random number generator), and we performed three to six independent runs of the simulations. Thus, in each data set, we have at least $8,191 \times 3 = 24,573$ replicate data points for each combination of the factors.

**Data availability.** All MATLAB simulation codes used in this study are available in the GitHub repository (https://github.com/debudutta1/noisy-cell-growth).

## ACKNOWLEDGMENTS

This research received no specific grant from any funding agency in the public, commercial, or not-for-profit sectors.

D.D.'s Ph.D. stipend was supported by the Department of Science & Technology, Innovation in Science Pursuit for Inspired Research (DST-INSPIRE) Fellowship (IF160091), Government of India.

We are grateful to Christopher Marx for valuable discussions and feedback. We also acknowledge the ICTS-ICTP Quantitative Systems Biology Schools 2015 and 2017 and the ICTS Bangalore School on Population Genetics and Evolution 2020 for the learning opportunity and exposure.

D.D. and S.S. conceptualized the study. D.D. and S.S. developed the theory. D.D. performed the simulations. D.D. wrote the manuscript with support from S.S.

We declare that we have no competing interests.

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
