## [Reviewer comments · mSystems]

Cell growth model with stochastic gene expression helps understand the growth advantage of metabolic exchange and auxotrophy

Dibyendu Dutta and Supreet Saini

Corresponding Author(s): Dibyendu Dutta, Indian Institute of Technology Bombay

Review Timeline:

Submission Date:	April 13, 2021
Editorial Decision:	May 25, 2021
Revision Received:	June 10, 2021
Editorial Decision:	July 6, 2021
Revision Received:	July 8, 2021
Accepted:	July 9, 2021

Editor: James Stegen

Reviewer(s): The reviewers have opted to remain anonymous.

Transaction Report:

DOI: <https://doi.org/10.1128/mSystems.00448-21>

May 25, 2021

Mr. Dibyendu Dutta
Indian Institute of Technology Bombay
Chemical Engineering
IIT Area, Powai
Mumbai, Maharashtra 400076
India

Re: mSystems00448-21 (Cell growth model with stochastic gene expression helps understand the growth advantage of metabolic exchange and auxotrophy)

Dear Mr. Dibyendu Dutta:

Thank you for submitting your manuscript to mSystems. We have completed our review and I am pleased to inform you that, in principle, we expect to accept it for publication in mSystems. However, acceptance will not be final until you have adequately addressed the reviewer comments.

Both reviewers raise important issues that need to be addressed in the manuscript itself, not just in the response letter. As Reviewer 4 notes, there were thorough responses in the letter to reviewers, but not as much done to the actual manuscript. Substantive edits are needed to the manuscript itself in light of the reviewers' comments. While I am providing a decision of 'minor revision' the suggested changes and issues raised are quite important and I will look carefully at the responses and the degree to which they are reflected in the manuscript.

Preparing Revision Guidelines

For complete guidelines on revision requirements, please see the Instructions to Authors at <https://msystems.asm.org/sites/default/files/additional-assets/mSys-ITA.pdf>. **Submissions of a paper that does not conform to mSystems guidelines will delay acceptance of your**

manuscript.

Sincerely,

James Stegen

Editor, mSystems

Journals Department
Reviewer comments:

Reviewer #3 (Comments for the Author):

The authors have continued to improve their manuscript. A few minor comments should be addressed.

Line 17: 'herd protection through biofilm structures' is out of place in the abstract as it is not a major theme of the paper. Please remove.

Line 52: 'Unlike pure cultures grown in the laboratory' please modify the current statement to something like the suggestion here.

Based on the Pande study and their reported AA exchanges, how many 'limiting resources' would be estimated for *E. coli*? Crossfeeding the appropriate AA increased fitness/growth rate so these AAs would fall under the definition of limiting resource presented in the current study. How does this number of limiting metabolites and the reported fitness/growth advantages in Pande correlate with the predicted growth advantages presented in Figure 4 of this study?

Line 499: The document argues there would be no resource savings with the proposed mechanism. This is not accurate if the enzyme flux can be described by Michaelis-Menten kinetics. Symmetric reciprocal cross feeding would result in a resource savings (aggregate metabolite and protein investment) for both cells because doubling the flux to produce resource A does not require a doubling of cellular investment, there is a nonlinear relationship between flux and investment. Higher fluxes do not require proportionally higher resource investments as shown by a couple

different groups doi: <https://doi.org/10.1101/128009>; <https://doi.org/10.1016/B978-0-444-63475-7.00015-7>, thus resulting in a division of labor resource saving that could be directed toward growth. This advantage would be independent of the exchanged resource and the genetic background of the considered cells.

Reviewer #4 (Comments for the Author):

Overall, the updates seem to be fairly minimal from the prior version of the manuscript but the responses to review concerns were thorough.

In response to the authors' assertion that cooperative growth rates are higher than prototrophic wild-type strains, the authors' interpretation of data from Harcombe (2010) and Pande (2014) are not correct. Those reports measure cooperator frequency (Harcombe) or relative fitness (Pande) after 24-48 hrs of growth. Neither data set directly indicates that the growth rate of cooperating auxotrophic strains of *E. coli* are higher than wild-type strains. Pande (2014) states cooperators grow faster but does not support this statement with any data in the paper. While it is possible growth rate is the cause of increased fitness, later work from Kost's lab indicates there is a role for nanotube formation between these cross-feeding *E. coli* suggesting an alternative mechanism to simply growing at a faster rate than wild-type. All this to say, growth rate is a component of competitive fitness in batch culture but it is not the only dimension.

Additionally, there is a statement in the introduction that sets the tone for this study that I don't think is accurate. Line 66-68 states: "However, to emerge as the dominant strategy in any ecological niche, cooperative consortiums must grow faster than other free-living species." Many (if not most) cooperative microbes, even those cited in 1-5, are in complex ecological niches that allow a high degree of resource partitioning. That said, these cooperative systems don't necessarily need to "grow faster than other free-living species". These cooperative relationships simply need to allow these microbes to access a new niche or remain at some frequency in an otherwise diverse, competitive environment. With this in mind, I am concerned that this model is really just looking for parameters that fit a condition that is not really realized in natural environments.

On going through this work again, I am finding data a bit challenging to reconcile. In particular, when considering the benefits of cooperative growth. Figure 3h shows that cells with a secretion ratio of 0.5 (as used in the cooperation simulations) have a ~20-40% reduction in growth rate. How can this not end up being reflected in the cooperative form of the model. Is it just that increasing the secreted metabolite by 2X (at no apparent cost to the producer) effectively reduces the costs associated with the 0.5 secretion ratio?

Response to Reviewer Comments

We would like to take this opportunity to thank the reviewers for their thorough evaluation of our revised manuscript. We have attempted to address the reviewers' comments, and have revised the manuscript thoroughly to ensure that readers are conveyed a clear and balanced narrative.

Moreover, to adhere to the journal's word limit, we have now moved the whole of Results section 6 to the Supplement S12. We have also restructured the Discussion section and incorporated summaries of discussions with the reviewers. Lastly, we have also made some changes to the introduction and results to aid readability and reduce wordiness.

We hope that the current version addresses all the concerns of the reviewers and meets the scientific standards of the journal.

Please find the Supplement associated with the manuscript at https://raw.githubusercontent.com/debudutta1/noisy-cell-growth/main/Supplementary_Results_Discussions.pdf

Please find below our specific responses addressing each of the concerns of the reviewers. The reviewer comments are coloured **brown**, while our responses are coloured **black** and indented.

Reviewer #3 (Comments for the Author):

The authors have continued to improve their manuscript. A few minor comments should be addressed.

Thank you.

Line 17: 'herd protection through biofilm structures' is out of place in the abstract as it is not a major theme of the paper. Please remove.

We have removed the reference, and revised the abstract. Line 17.

Line 52: 'Unlike pure cultures grown in the laboratory' please modify the current statement to something like the suggestion here.

Sadly, the suggestion text was missing in the responses sent to us. We have nevertheless re-written the sentence. (Line 52).

Based on the Pande study and their reported AA exchanges, how many 'limiting resources' would be estimated for E. coli? Crossfeeding the appropriate AA increased fitness/growth rate so these AAs would fall under the definition of limiting resource presented in the current study. How does this number of limiting metabolites and the reported fitness/growth advantages in Pande correlate with the predicted growth advantages presented in Figure 4 of this study?

Normally a “limiting resource” refers to any resource whose shortfall can limit growth. Thus, all precursor metabolites considered in the biomass objective function of a genome-scale Metabolic Flux model, are eligible. However, this is true only for the optimal steady-state flux distribution. In our stochastic model, the metabolites whose unavailability delays cell division most often (as an emergent outcome of the enzyme kinetic parameters, gene expression parameters and metabolic demand) are considered “limiting”. We expect such “kinetic bottlenecks” may be metabolites with very high demand, or with many upstream flux shunts, or produced by biosynthetic enzymes with large gene expression noise. At present, we are unable to identify them, but by scaling-up our abstract model to the genome-scale and inserting experimentally observed gene-expression parameters may help identify them in the future. (This has been added to the revised Discussion section Lines 421-430).

Moreover, in our model, if any of the multiple stochastic biosynthetic pathways require the same time on average to produce the requirement threshold quantity of metabolites, we consider them to be “bottleneck metabolites” of the same strength (p). It is while comparing such metabolites that we obtain the Fig – 4d. In case, the mean time required for production is lower, the metabolite acts as a bottleneck but of lower strength (q). The growth differences associated with such auxotrophies are discussed in Supplement S10.

In the Pande study, different synthetic auxotrophs with different overexpression mutations must lead to different p “limiting metabolites”, based on our definition. Thus, it is not straightforward to guess the number p in these different genetic backgrounds.

Line 499: The document argues there would be no resource savings with the proposed mechanism. This is not accurate if the enzyme flux can be described by Michaelis-Menten kinetics. Symmetric reciprocal cross feeding would result in a resource savings (aggregate metabolite and protein investment) for both cells because doubling the flux to produce resource A does not require a doubling of cellular investment, there is a nonlinear relationship between flux and investment. Higher fluxes do not require proportionally higher resource investments as shown by a couple different groups doi: <https://doi.org/10.1101/128009>; <https://doi.org/10.1016/B978-0-444-63475-7.00015-7>, thus resulting in a division of labor resource saving that could be directed toward growth. This advantage would be independent of the exchanged resource and the genetic background of the considered cells.

Considering classical Michaelis-Menten kinetics, investment in enzymes varies linearly with flux, however, it is indeed nonlinear with substrate. If the substrate flux was originally very low, then one can increase the Substrate flux to the pathway and increase the amount of metabolite A produced. However, in case Substrate flux was already high, there would be no further increase in metabolite flux. We demonstrate this in our simulations, when we vary S_{in} in our model (Fig – 2d), without a change in the enzyme. However, specifically redistributing substrate flux

to one substrate limited pathway will cause an equivalent shortfall in the pathway from which the flux is withdrawn, and will not improve growth.

In Figure – 4b, for low S_{in} we see the auxotrophs grow much better (than prototrophs) because the substrate flux freed up by the loss of one pathway is redistributed equally among the remaining pathways (b in figure below), which increases metabolite production flux without any change in enzyme production.

Similarly, during symmetric reciprocal cross-feeding between complementary auxotrophs, the saved substrate flux will be utilized by all pathways using the same

substrate, unless some form of regulation is used to channel this saved flux to the cross-fed metabolite pathway (c in figure below). We use enzyme over-expression to channel this substrate flux to the cross-fed metabolite pathway (d in figure below), which utilizes the saved protein resource from auxotrophy. Alternative regulatory schemes may be possible, but we do not consider them in our analysis.

Given this scheme considered in our model, the benefits of saved resources would be lost (judging from a steady state perspective – See Supplement S5), and hence our assertion.

We have now revised the Discussion section and the entire paragraph that contained the sentence discussed is removed. The same points are however also discussed in results section 5 (Lines 361-376).

Reviewer #4 (Comments for the Author):

Overall, the updates seem to be fairly minimal from the prior version of the manuscript but the responses to review concerns were thorough.

Thank you, we have sincerely attempted to address all the raised concerns, and incorporate them in the main text or supplement wherever we felt necessary.

In response to the authors' assertion that cooperative growth rates are higher than prototrophic wild-type strains, the authors' interpretation of data from Harcombe (2010) and Pande (2014) are not correct. Those reports measure cooperator frequency (Harcombe) or relative fitness (Pande) after 24-48 hrs of growth. Neither data set directly indicates that the growth rate of cooperating auxotrophic strains of *E. coli* are higher than wild-type strains. Pande (2014) states cooperators grow faster but does not support this statement with any data in the paper.

Pander *et al.* 2014 in the supplement to their manuscript provide the complete dataset of empirical growth rates for all tested complementary auxotrophic pairs in Figure S2 (https://static-content.springer.com/esm/art%3A10.1038%2Fisme.2013.211/MediaObjects/41396_2014_BFisme.2013.211_MOESM19_ESM.doc).

Figure S2 shows 43 of 54 combinations of the synthetic complementary auxotrophs growing better than the wildtype prototroph.

While it is possible growth rate is the cause of increased fitness, later work from Kost's lab indicates there is a role for nanotube formation between these cross-feeding *E. coli* suggesting an alternative mechanism to simply growing at a faster rate than wild-type. All this to say, growth rate is a component of competitive fitness in batch culture but it is not the only dimension.

While nanotube formation addresses the question of direct supply without loss of the partner's production, it still does not explain how the complementary pair of auxotrophs can exhibit a growth rate that exceeds the ancestral prototroph's (such as in Pande *et al.* 2014), even for the case of symmetric exchange.

We agree that growth rate is only a component of overall microbial fitness, but it is the major metric that is measured in batch cultures, and reported as fitness. Factors other than growth rate is beyond the scope of our model in its present state.

We believe that our model uncovers nuances of bacterial growth that remain hidden because our usual conceptual framework relies on a steady-state picture.

Additionally, there is a statement in the introduction that sets the tone for this study that I don't think is accurate. Line 66-68 states: "However, to emerge as the dominant strategy in any ecological niche, cooperative consortiums must grow faster than other free-living species." Many (if not most) cooperative microbes, even those cited in 1-5, are in complex ecological niches that allow a high degree of resource partitioning. That

said, these cooperative systems don't necessarily need to "grow faster than other free-living species". These cooperative relationships simply need to allow these microbes to access a new niche or remain at some frequency in an otherwise diverse, competitive environment. With this in mind, I am concerned that this model is really just looking for parameters that fit a condition that is not really realized in natural environments.

Indeed, it is unnecessary for the cooperative system to grow faster than other free-living species, and it should be sufficient to access an exclusive niche and persist at some frequency in the environment. However, in the course of evolution it is likely that multiple species (cooperative and free-living) competed for each of the resource partitions and only those with the best community productivity and consequent growth characteristics, ultimately remained. Few studies have attempted to investigate the assembly rules for consortiums and how these may evolve over time (Higgins et al. 2017), and found that only the most competitive strains remain when multiple strains of similar nature are mixed together (Meroz et al. 2021).

It is in this context of evolution towards cooperation via competition, that we make the statement about dominant strategy. Our framework attempts to describe and capture growth effects that maybe missed by existing steady-state models, which would have important implication for simulation of evolution of cooperation between microbes.

Nevertheless, we have tweaked the sentence: "However, to perdure in an ecological niche, cooperative consortiums must demonstrate high community productivity and growth compared to other competing (free-living or cooperating) species". (Lines 65-67).

Higgins, Logan, Jonathan Friedman, Hao Shen, and Jeff Gore. 2017. "Co-Occurring Soil Bacteria Exhibit a Robust Competitive Hierarchy and Lack of Non-Transitive Interactions." *BioRxiv*, August, 175737. <https://doi.org/10.1101/175737>.

Meroz, Nittay, Nesli Tovi, Yael Sorokin, and Jonathan Friedman. 2021. "Community

Composition of Microbial Microcosms Follows Simple Assembly Rules at Evolutionary Timescales." *Nature Communications* 12 (1): 2891.
<https://doi.org/10.1038/s41467-021-23247-0>.

On going through this work again, I am finding data a bit challenging to reconcile. In particular, when considering the benefits of cooperative growth. Figure 3h shows that cells with a secretion ratio of 0.5 (as used in the cooperation simulations) have a ~20-40% reduction in growth rate. How can this not end up being reflected in the cooperative form of the model. Is it just that increasing the secreted metabolite by 2X (at no apparent cost to the producer) effectively reduces the costs associated with the 0.5 secretion ratio?

In short, yes.

Please note that Figure – 3h describes the growth effect of secreting one limiting metabolite in case of a prototroph, when there is no overexpression. Secretion decreases the rate at which internal requirement can be met and hence division time increases and growth rate decreases.

For the cooperation simulation, we use a secretion ratio of 0.5, but simultaneously we also overproduce the enzyme (implementing a doubling of stochastic production flux) for the pathway (Fig – 5a). Thus, effectively the cooperating cell makes twice the original amount (twice the cellular requirement) and secretes half of it for the partner cell. Hence, both cooperating cells get the required amount of metabolite in overall lesser time, and so grow faster (Fig – 5b).

The protein resource saved due to auxotrophy (i.e. non-production of enzymes for the lost pathway) is used to overproduce the biosynthetic enzymes of the cross-fed metabolite. Since, no additional resources are expended there is no deleterious effect on growth.

July 6, 2021

Mr. Dibyendu Dutta
Indian Institute of Technology Bombay
Chemical Engineering
IIT Area, Powai
Mumbai, Maharashtra 400076
India

Re: mSystems00448-21R1 (Cell growth model with stochastic gene expression helps understand the growth advantage of metabolic exchange and auxotrophy)

Dear Mr. Dibyendu Dutta:

Thank you for submitting your manuscript to mSystems. We have completed our review and I am pleased to inform you that, in principle, we expect to accept it for publication in mSystems. However, acceptance will not be final until you have adequately addressed the reviewer comments.

Both reviewers provided some additional, minor items to be addressed. Reviewer 3 indicates some dissatisfaction with the study due to being purely theoretical. This is a valid concern that does not preclude publication, but that does warrant the inclusion of an additional paragraph in the Discussion highlighting the limitations of the study, along the lines of Reviewer 3's comments. Reviewer 4 also has some important comments that need to be addressed. In addition, I believe the variable names are flipped around the equation below line 198. Please check that carefully. Lastly, I could find no statement of code or data availability. It is important to make your computer code available, along with any input data, and making model outputs available would be even better. Please indicate how readers can access these materials, which are needed to reproduce your study and analyses.

Preparing Revision Guidelines

For complete guidelines on revision requirements for your article type, please see the journal Article Types requirement at <https://journals.asm.org/journal/mSystems/article-types>. **Submissions of a paper that does not conform to mSystems guidelines will delay acceptance of your manuscript.**

Sincerely,

James Stegen

Editor, mSystems

Journals Department
Reviewer comments:

Reviewer #3 (Comments for the Author):

The study applies gene noise to a mathematical model to recreate emergent properties of consortia interactions, namely enhanced growth rates. The math predicts enhanced growth rate under certain conditions however, this mechanism does not likely represent a major, biologically relevant mechanism. The research is purely theoretical and the authors are not able to quantitatively use their model to quantify growth enhancements seen in previous work, e.g. Pande et al. The provided explanations for this short coming, an unknown number of limiting metabolites, are disappointing highlighting weaknesses to the theory. Explicitly what experimental data would be required to test the proposed theory?

The work will contribute to the growing list of various mathematical theories being applied to study consortia and their properties even if the mechanism is not likely a major driver of the properties.

Reviewer #4 (Comments for the Author):

Overall, the authors have worked hard to make a more concise and clear manuscript. It is a nice study and will hopefully serve as a platform for further refinement of the modeling approach. I only

have a couple items to consider.

The authors indicated in their response that Pande et al., 2014 (Reference 11) reported empirical growth rate data in Figure S2 of their published work. This is not correct. The data reported in that paper (specifically that figure) is a Malthusian parameter corrected for 24 hrs of growth. Effectively, this is just a log ratio of colony forming units and not a growth rate. While growth rate and fitness (or productivity) can be (and are often) closely linked in batch growth, this doesn't necessarily have to be true. I respect that Pande and coworkers state that growth rate increased in their cooperative mutants but that was not what was demonstrated by the data in their published work. What I would argue their data shows more clearly is an increase in productivity. Regardless, I would encourage amending various statements to reflect that Pande and coworkers observed increased fitness.

Line 364-366 states: "How must cellular resources be balanced between these processes to achieve a net positive outcome is not understood, even for a simple symmetric exchange." There have been some insights into this using dynamic FBA frameworks like what is used in COMETS (Harcombe et al, Cell Reports 2014). The COMETS dFBA framework was able to capture behavior of a cooperative system based on metabolite exchange.

In Figure 4b and Figure 5b, what is the feed rate for the missing metabolite in the cases of the prototroph and auxotroph? How are these being compared (i.e. are the simulations run at the same feed rate)? This should be clarified.

Response to Reviewer Comments

We would like to take this opportunity to thank the reviewers for their thorough evaluation of our revised manuscript. We have attempted to address the reviewers' comments, and have made necessary changes in the manuscript to ensure that readers are conveyed a clear and balanced narrative.

We hope that the current version addresses all the concerns of the reviewers and meets the scientific standards of the journal.

Please find the Supplement associated with the manuscript at

https://raw.githubusercontent.com/debudutta1/noisy-cell-growth/main/Supplementary_Results_Discussions.pdf

Please find below our specific responses addressing each of the concerns of the reviewers. The reviewer comments are coloured **brown**, while our responses are coloured **black** and indented.

Reviewer #3 (Comments for the Author):

The study applies gene noise to a mathematical model to recreate emergent properties of consortia interactions, namely enhanced growth rates. The math predicts enhanced growth rate under certain conditions however, this mechanism does not likely represent a major, biologically relevant mechanism. The research is purely theoretical and the authors are not able to quantitatively use their model to quantify growth enhancements seen in previous work, e.g. Pande et al. The provided explanations for this short coming, an unknown number of limiting metabolites, are disappointing highlighting weaknesses to the theory. Explicitly what experimental data would be required to test the proposed theory?

Thank you for your rigorous reviews that have helped us improve the manuscript a great deal.

We are greatly aware of the limitation of this study being purely theoretical as presented in this manuscript. We had originally set up a collaboration under the Newton-Bhabha PhD Placement program and intended to obtain distributions of cell generation times for prototrophic and auxotrophic *E. coli*, when fed metabolites like amino acids at various rates. However, due to the pandemic we were unable to execute this plan.

A match between the experimental distributions of generation times and our theoretical model would provide much support and give clues about the active number of bottlenecks in the cell. These experimentally obtained datasets can be used with metabolic flux models to identify plausible bottleneck metabolites. Based on the cell's growth responses to the supply of different classes of metabolites, we think it is possible to identify the "bottleneck" metabolites as we defined in our model.

We have modified the discussion section to admit this limitation more explicitly and have also included a summary of our experimental approach to identify these bottlenecks. (Lines 430-438).

The work will contribute to the growing list of various mathematical theories being applied to study consortia and their properties even if the mechanism is not likely a major driver of the properties.

Thank you for your encouraging assessment.

Reviewer #4 (Comments for the Author):

Overall, the authors have worked hard to make a more concise and clear manuscript. It is a nice study and will hopefully serve as a platform for further refinement of the modeling approach. I only have a couple items to consider.

Thank you for the encouraging comments.

The authors indicated in their response that Pande et al., 2014 (Reference 11) reported empirical growth rate data in Figure S2 of their published work. This is not correct. The data reported in that paper (specifically that figure) is a Malthusian parameter corrected for 24 hrs of growth. Effectively, this is just a log ratio of colony forming units and not a growth rate. While growth rate and fitness (or productivity) can be (and are often) closely linked in batch growth, this doesn't necessarily have to be true. I respect that Pande and coworkers state that growth rate increased in their cooperative mutants but that was not what was demonstrated by the data in their published work. What I would argue their data shows more clearly is an increase in productivity. Regardless, I would encourage amending various statements to reflect that Pande and coworkers observed increased fitness.

We would like to respectfully disagree with your assessment that the Malthusian parameter does not represent the growth rate.

Malthusian growth model	Growth rate as slope of log OD vs t
The Malthusian growth model or the simple exponential growth model states: $P_t = P_0 e^{Mt}$ $\Rightarrow M = \frac{1}{t - 0} \log\left(\frac{P_t}{P_0}\right)$ P_t represents the population at time t, and P_0 the starting population. M is the Malthusian growth parameter and t represents the time.	Growth rate, as commonly measuring using the slope of the line on the log(OD600) vs time plot: $\mu = \frac{\log(OD_f) - \log(OD_i)}{t_f - t_i}$ $\Rightarrow \mu = \frac{1}{t_f - t_i} \log\left(\frac{OD_f}{OD_i}\right)$ μ represents the growth rate, OD_f is the final OD reading measured at time t_f, OD_i is the initial OD reading measured at time t_i.

--	--

Please note that the above two equations are mathematically the same, thus the Malthusian growth parameter M and μ are equivalent.

The only difference is in how the bacterial population is measured, while Pande *et al.* chose to use CFU instead of the usual OD measurements. These differences in scale however, does not matter since the measure of concern is the ratio of final population size to initial population size.

Line 364-366 states: "How must cellular resources be balanced between these processes to achieve a net positive outcome is not understood, even for a simple symmetric exchange." There have been some insights into this using dynamic FBA frameworks like what is used in COMETS (Harcombe et al, Cell Reports 2014). The COMETS dFBA framework was able to capture behavior of a cooperative system based on metabolite exchange.

Thank you for pointing out the relevance of this paper in this context, we have modified the sentence to include the citation.

Lines 365-367: "How must cellular resources be balanced between these processes to achieve a net positive outcome is not completely understood, even for a simple symmetric exchange. (Harcombe et al, Cell Reports 2014 provides some insights into the process)."

In Figure 4b and Figure 5b, what is the feed rate for the missing metabolite in the cases of the prototroph and auxotroph? How are these being compared (i.e. are the simulations run at the same feed rate)? This should be clarified.

In Figure 4b and Figure 5b, we plot the growth of auxotrophs at various feed rates (coloured solid lines) while also varying the substrate import (X-axis). The feed rates represented by the coloured lines are in the legend.

The prototroph growth represented by the dotted lines in these figures is the data from Figure 2d. We plot these to help readers compare the change in growth pattern.

Thank you for pointing this out the ambiguity, we have now modified the figure legends (Lines 860-866) and the main text (Lines 315-316) to clarify these points.

July 9, 2021

Mr. Dibyendu Dutta
Indian Institute of Technology Bombay
Chemical Engineering
IIT Area, Powai
Mumbai, Maharashtra 400076
India

Re: mSystems00448-21R2 (Cell growth model with stochastic gene expression helps understand the growth advantage of metabolic exchange and auxotrophy)

Dear Mr. Dibyendu Dutta:

Thank you for addressing the reviewer comments and my suggestions. I found your argument regarding Malthusian growth particularly well done. In addition, thank you for sticking with the review process, I realize there were many rounds. Congratulations on having the tenacity to see it through and producing a valuable contribution.

Your manuscript has been accepted, and I am forwarding it to the ASM Journals Department for publication. For your reference, ASM Journals' address is given below. Before it can be scheduled for publication, your manuscript will be checked by the mSystems senior production editor, Ellie Ghatineh, to make sure that all elements meet the technical requirements for publication. She will contact you if anything needs to be revised before copyediting and production can begin. Otherwise, you will be notified when your proofs are ready to be viewed.

As an open-access publication, mSystems receives no financial support from paid subscriptions and depends on authors' prompt payment of publication fees as soon as their articles are accepted. =

Publication Fees:

- Minimum resolution of 1280 x 720

- .mov or .mp4. video format
- Provide video in the highest quality possible, but do not exceed 1080p
- Provide a still/profile picture that is 640 (w) x 720 (h) max
- Provide the script that was used

We recognize that the video files can become quite large, and so to avoid quality loss ASM suggests sending the video file via <https://www.wetransfer.com/>. When you have a final version of the video and the still ready to share, please send it to Ellie Ghatineh at eghatineh@asmusa.org.

Sincerely,

James Stegen
Editor, mSystems

Journals Department
Phone: 1-202-942-9338